# Prevalence and anatomical significance of the persistent median artery: A cadaveric study

**Connor Ellis**[ID]*, **Drew Thibault**[ID], **Josh Lencke**[ID], **Laurieanne D. Hemric**

Department of Anatomical Sciences, Liberty University College of Osteopathic Medicine, Lynchburg, Virginia, United States of America

☉ All authors contributed equally to this work.

\* chellis1@liberty.edu

## Abstract

### Purpose

The median artery is the initial, yet transient, vascular supply to the palm originating from the brachial artery during development. This artery generally regresses as radial and ulnar arterial supply become prominent; however, this regression does not always occur. This study seeks to illuminate the prevalence of the persistent median artery in the adult population, as well as describe the vascular consequences secondary to the presence of this artery.

### Methods

Antebrachial and palmar dissections were performed on 56 adult cadavers (112 upper extremities). Prevalence of palmar and antebrachial-type persistent median artery were recorded separately. Arterial diameters of radial and ulnar arteries were measured and statistically examined with and without the co-presence of a persistent median artery.

### Results

A persistent median artery was present in 43% of the upper limbs included in this study. 28% of the population were of the antebrachial-type, while 15% were of the palmar-type. The overall mean arterial diameters of the radial artery were 3.13 mm (± 0.57 mm; $n = 52$) without any persistent median artery present, compared to 3.09 mm (± 0.56 mm; $n = 34$) with persistent median artery co-presence. The mean diameters for the ulnar artery were 2.75 mm (± 0.53 mm; $n = 53$) without co-presence of persistent median artery and 2.86 mm (± 0.65 mm; $n = 36$) with co-presence.

### Conclusion

This and other studies have demonstrated the prevalence of this persisting embryologic artery to be quite high. An understanding of the characteristics and significance of this artery in the adult population is necessary to surgeons operating on the carpal tunnel or distal antebrachium.

**Data availability statement:** All relevant data are within the manuscript and its Supporting Information files.

**Funding:** The author(s) received no specific funding for this work.

**Competing interests:** All authors contributed equally to this work.

## Introduction

The median artery is an embryologic artery originating from the brachial artery around the 5th embryonic week. This artery is known to degenerate around the 8th week of development as it is replaced first by the ulnar, then the radial, arteries [1,2]. However, this regression does not always occur. The product of this incomplete degeneration is termed a persistent median artery (PMA), which may be present in either a palmar or an antebrachial pattern [3]. In the adult, these arteries may be found branching from the brachial, ulnar, radial, anterior interosseous, or common interosseous artery, and running distally in the antebrachium within the epineurium of the median nerve [4]. The antebrachial pattern travels alongside the median nerve within the epineurium and terminates prior to the transverse carpal ligament (TCL), supplying the median nerve (Fig 1A). The palmar pattern will continue deep to the TCL and will participate in palmar vascularization. Further, the palmar variant may itself be present in several different patterns, terminating as an anastomosis to the superficial palmar arch or may even be the sole vascular supply to an individual digit (Fig 1B) [5]. Therefore, ligation or resection of this artery may cause unintentional digital hypoperfusion or avascular necrosis. Additionally, a correlation between a palmar-type PMA (pPMA) and both carpal tunnel syndrome and anterior interosseous syndrome has been postulated, though direct causation is still debated [6–8]. With an estimated 11–22% of all ER visits in the United States related to hand and wrist trauma, study of this anatomical region is of utmost importance [9]. This cadaveric study will have two main study objectives: (i) to ascertain the prevalence of a PMA in a cadaveric population, especially of the pPMA and (ii) to determine if there is a correlation between the presence of a pPMA and decreased diameters of radial and ulnar arteries. The mean diameters for the major antebrachial and palmar vessels will be reported, along with analyses demonstrating each artery's relationship to the others.

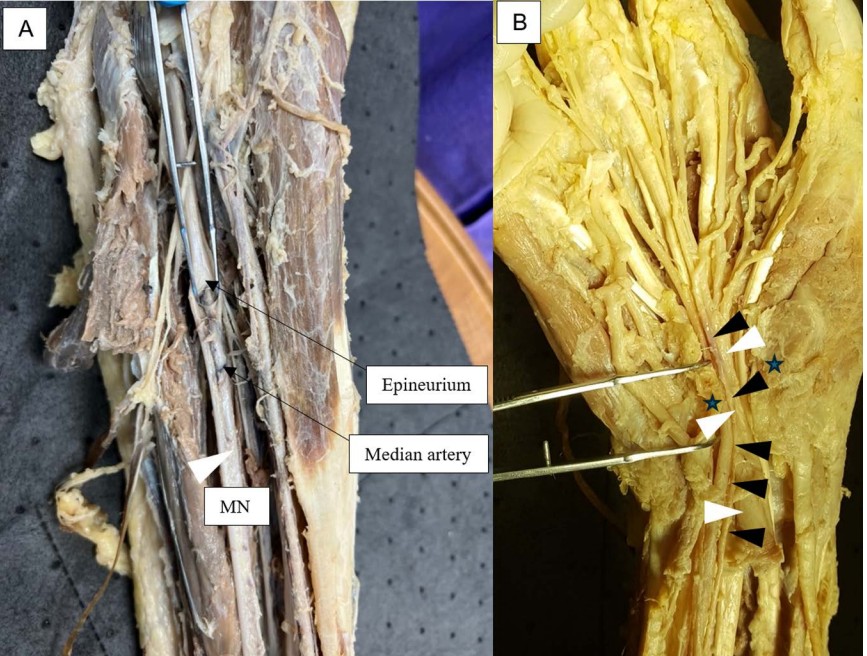

**Fig 1. A.** The persistent median artery continues within the epineurium of the median nerve (MN), often bisecting it (white arrowhead). **B**. The palmar-type persistent median artery (black arrowhead) continues through the carpal tunnel alongside the median nerve (white arrowhead), deep to the flexor retinaculum (star), and into the volar hand.

## Materials and methods

56 cadavers (112 upper extremities) acquired from the Virginia State Anatomical Program (https://www.vdh.virginia.gov/medical-examiner/vsap/) for undergraduate and medical school anatomical laboratories were included in this study and were examined over the period of 07/27/2022—12/27/2023. The study population was aged between 62 and 100 years of age and were chosen for study without clinical requirements, that is, the presence of hypertension, smoking history, diabetes, carpal tunnel syndrome, anterior interosseous syndrome, and other corresponding pathologies were neither screened for nor will be reported here [10,11]. Cadavers were initially dissected by undergraduate or medical students, depending on which lab they were acquired from. Those cadavers that required additional dissection for full visualization of the anatomy in this investigation were dissected by the researchers of this study. Dissection was carried out in the following fashion: (i) antebrachial skin was resected and antebrachial musculature was reflected with the intention of viewing the terminal brachial artery and the proximal radial, ulnar, common interosseous, and anterior interosseous arteries; (ii) the median nerve was exposed, and the relationship between the median nerve and the surrounding vasculature was examined with the intention of identifying the presence of an artery branching from one of the aforementioned arteries and running within the epineurium of the median nerve; (iii) dissection was continued into the palm with removal of overlying integument, palmar fascia, and division of the transverse carpal ligament; (iv) pPMA were followed from the antebrachial origin through the carpal tunnel and to their terminations either on the superficial palmar arch (SPA), radialis indicis, or as a collection of common palmar digital arteries; (v) finally, the adventitial fascia was removed from the aforementioned arteries to allow for uniform external diameter measurement.

Several characteristics have been used in the definition of a PMA when discovered in an adult individual; these authors defined PMA as an artery that both (i) originates from the brachial, radial, ulnar, common interosseous, or anterior interosseous artery, and (ii) travels within the epineurium of the median nerve to either the distal antebrachium or palm [1,3,12,13]. Antebrachial-type PMA (aPMA) were defined as those that terminated within the distal antebrachium, while palmar-type PMA (pPMA) were defined as those PMA that continued into the palm deep to the transverse carpal ligament (Fig 1B)

Following dissection and exposure of the arteries in question, prevalence of each artery was recorded with emphasis on antebrachial-type and palmar-type PMA. Because a pPMA is an anatomical continuation of the antebrachial PMA, the authors did not include pPMA in the prevalence count of aPMA, and therefore did not include aPMA in the prevalence count of pPMA. Variant arterial laterality was also recorded for investigation on lateral preference and bilateral comparison.

When present, diameter of the radial, ulnar, pPMA, aPMA, SPA, and deep palmar arch (DPA) was recorded bilaterally. Arterial diameters were measured with an analogue caliper. Each artery was measured in triplicate by a single researcher, then these 3 measurements were averaged for the final recorded diameter. Arterial walls were examined without compression in order to preserve original luminal patency and therefore give the closest approximation to *in vivo* arterial diameter. Measurements were taken in the following manner: the radial artery was measured both at the level of the radial styloid process just proximal to the scaphoid, as well as dorsally within the first metacarpal space just distal to the extensor pollicis longus tendon crossing. The ulnar artery was measured just proximal to the pisiform, prior to the branch point of the deep palmar branch of the ulnar artery. The SPA was measured between the branch point of the 2nd and 3rd common palmar digital arteries; similarly, the DPA was measured between the branch point for the 2nd and 3rd palmar metacarpal arteries. PMA were measured 1 cm distal to the initial branch point proximally in the antebrachium; when it was

discovered that the PMA was palmar-type, this artery was measured again immediately distal to the proximal lip of the transverse carpal ligament.

Statistical analysis was conducted with JMP Pro 17 (JMP®, Version *17*. SAS Institute Inc., Cary, NC, 1989–2023). The level of significance was set at $p$ = 0.05. Radial, ulnar, SPA, and DPA arterial diameters were compared with their contemporary on the contralateral side. PMA diameters, however, were not compared with contralateral arterial presence due to the low sample size. Radial artery diameter was compared with ipsilateral ulnar artery diameter in an effort to describe each artery's relationship to the other [11,14,15]. The present authors postulated the presence of a PMA, specifically the palmar-type, may add to the vascular supply of the palm significantly enough to alter the average size of both radial and ulnar arteries. To investigate this hypothesis, *t*-tests were conducted comparing mean radial artery diameter without the co-presence of a pPMA with the mean radial arterial size with the co-presence of a pPMA. Further, the radial artery diameter was examined comparing the mean diameter without the presence of an aPMA with the radial artery diameter with the presence of an aPMA. The ulnar artery was examined in similar fashion: mean diameters of the ulnar artery without the presence of a pPMA were compared to those with a pPMA, and again the mean diameters of the ulnar artery without any PMA were compared to those with an aPMA. These diameter analyses were run with the total included arteries, as well as with each laterality separately. Single tail *t*-tests were run when a trend was noticed or expected: for example, single tail analysis was conducted for the change of RA, UA, and SPA with and without the presence of a PMA to determine if a trend was indeed occurring.

Linear regression was conducted in an attempt to discover predictive capacity of the diameter of an individual pPMA with the diameter of its corresponding ipsilateral radial artery. This analysis was also completed for the pPMA and its corresponding ipsilateral ulnar artery. These examinations were repeated comparing aPMA with their corresponding ipsilateral radial and ulnar arteries. Each linear regression was completed with both unilateral data and combined bilateral data. Linear regression was also run comparing unilateral radial, ulnar, SPA, and DPA diameters with the contralateral corresponding artery to examine for arterial diameter similarity between the extremities.

## Results

Of the 112 upper extremities initially dissected, only 89 extremities were suitable for PMA analysis. Of these, 43% ($n$ = 38) had a PMA present; further, 15% ($n$ = 13) were palmar-type, while 28% ($n$ = 25) were antebrachial-type (Fig 2).

Mean arterial diameter was recorded for suitable RA ($n$ = 87), UA ($n$ = 90), SPA ($n$ = 81), DPA ($n$ = 89), aPMA ($n$ = 25), and pPMA ($n$ = 13). Further, a single tailed *t*-test demonstrated that the radial artery was consistently and significantly larger than the ulnar artery ($p$ = 0.0003). We had expected similar bilateral dimensions when measuring the same artery in right and left limbs. Interestingly, regression analysis demonstrated no correlation between the arterial diameter of either extremity with its contralateral counterpart (Table 1). Additionally, it was noticed that neither laterality examined was consistently larger than its contralateral counterpart.

The external diameters of the RA, UA, SPA, and DPA were examined for change in the presence of a PMA (S1 Table). We hypothesized that an additional arterial supply within the antebrachium or palm may be associated with a decrease in external diameter of the RA and UA; additionally, there may be an associated increase in the diameter of the SPA due to the common anastomosis between the pPMA and the SPA. The RA diameter trended smaller with the presence of a PMA, though this was non-significant ($p$ = 0.82) (Fig 3A). The UA diameter

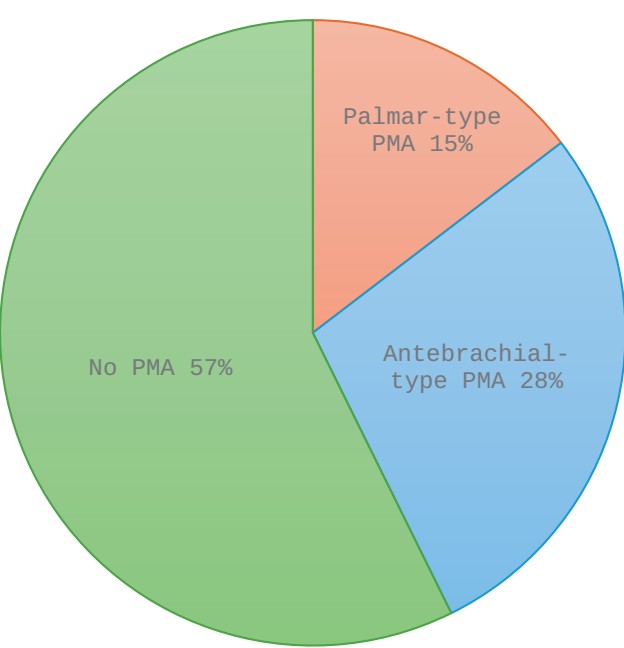

**Fig 2. Prevalence of the persistent median artery (PMA) in the study population.** Out of the 89 limbs examined for presence of the PMA, 57% (n = 51) did not have a PMA, and 43% (n = 38) did have a PMA, with 15% (n = 13) of the palmar-type, while 28% (n = 25) were of the antebrachial-type.

**Table 1. Comparison of arterial diameters in the upper limbs of cadaveric subjects.**

| Artery | Both limbs | Left | Right | Left v. right regression |
|---|---|---|---|---|
| **Radial** | 3.12 ± 0.56 mm (n = 87) | 3.09 ± 0.53 mm (n = 44) | 3.15 ± 0.59 mm (n = 43) | $r^2$ = 0.098 |
| **Ulnar** | 2.79 ± 0.58 mm (n = 90) | of 2.81 ± 0.64 (n = 47) | 2.78 ± 0.52 mm (n = 43) | $r^2$ = 0.005 |
| **Superficial palmar arch** | 2.07 ± 0.62 mm (n = 81) | 2.06 ± 0.59 mm (n = 42) | 2.08 ± 0.65 mm (n = 39) | $r^2$ = 0.062 |
| **Deep palmar arch** | 1.52 ± 0.47 (n = 89) | 1.60 ± 0.49 mm (n = 47) | 1.44 ± 0.44 mm (n = 42) | $r^2$ = 0.006 |
| **Palmar-type PMA** | 1.40 ± 0.58 mm (n = 13) | 1.55 ± 0.65 mm (n = 8) | 1.32 ± 0.36 mm (n = 5) | – |
| **Antebrachial-type PMA** | 1.35 ± 0.43 mm (n = 25) | 1.26 ± 0.50 mm (n = 12) | 1.44 ± 0.34 mm (n = 13) | – |

trended *larger* with the presence of a PMA, and this peculiarity rapidly approached significance ($p$ = 0.08) (Fig 3B). Of further interest, the presence of a pPMA was found to increase the mean diameter of the SPA ($p$ = 0.03) (Fig 3C) but had no correlated change in the DPA ($p$ = 0.16) (Fig 3D).

No form of the PMA had any measurable correlation on the co-present vascular diameters (S2 Table).

## Discussion

This study began as a cadaveric-dissection based report on pPMA prevalence and quickly morphed into a study examining the effect this embryologic arterial remnant has on existing

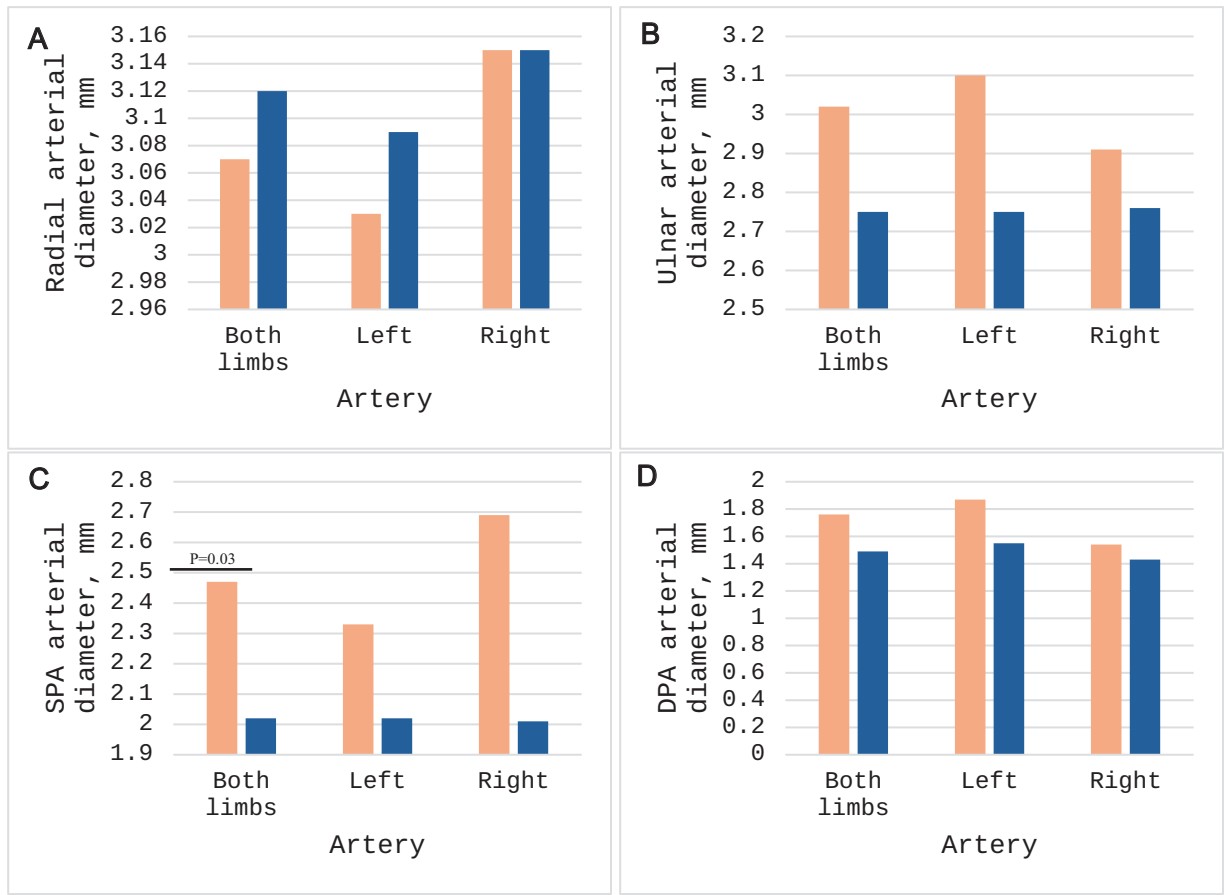

**Fig 3. Comparisons of antebrachial arterial diameters in the cases of presence (orange) and absence (blue) of persistent median artery (PMA); significant values labeled.**

antebrachial and palmar vasculature. After examining the high prevalence of the PMA in our medical and undergraduate laboratories, it was postulated that the presence of this additional artery branching from a brachial artery derivative might influence the size of the other arteries of the palm. As mean antebrachial and palmar arterial diameters are not *commonly* reported in the literature, we decided to compare arterial sizes in cadavers who possessed a PMA with those who did not possess a PMA [14–18]. We compared each laterality to both further understand each artery's relationship to its contemporaries, as well as examine for arterial diameter consistency between the extremities. The main data this paper hopes to report, however, is the embryological-remnant arterial relationship to the main arteries in the antebrachium and palm. To the knowledge of the authors, this is the first study that examines the impact of the PMA diameter on surrounding antebrachial and palmar vasculature diameters.

Out of the 89 upper extremities examined for PMA prevalence, 43% had a PMA ($n$ = 38). Further, 15% ($n$ = 13) of the population had a pPMA, while 28% ($n$ = 25) had an aPMA. These data are consistent with previously reported prevalence data in the adult population [19]. In their meta-analysis of 10,394 subjects, Solewski et al. found a pPMA prevalence of 8.6% and an aPMA prevalence of 34.0%. Reported prevalence ranges from 0% (Umapathy et al., 2012, $n$ = 50 extremities) to 100% (Blunt 1959, $n$ = 6 extremities); however, these studies range drastically in demographics and number of samples included [20,21]. Further,

prevalence in different aged populations has been found to vary widely. For example, Aragão et al. found a total prevalence of 81.25% ($n = 32$), however, this study's population was of fetuses rather than adults [22]. In the present study, the prevalence of PMAs for each laterality was very similar: 45% ($n = 20$) on the left and 40% ($n = 18$) on the right. Further breakdown into palmar and antebrachial subtypes showed similar results: 18% ($n = 8$) of left extremities and 11% ($n = 5$) of right extremities had a pPMA, while 27% ($n = 12$) of left extremities and 29% ($n = 13$) of right extremities had an aPMA present. Again, these data appear very similar to those reported by Solewski et al., who found a pPMA to be present 10.6% ($n = 1378$) on the left side and 9.7% ($n = 1413$) on the right in the adult population [19].

This study also examined the relationship between the radial and ulnar arterial diameters. It has been reported that the ulnar artery is the larger of the two arteries [11,15,17]. However, others have reported the very opposite [14]. This study found the radial artery was significantly larger than the ulnar artery ($p = 0.0003$). Interestingly, when left and right RA, UA, SPA, and DPA were compared with one another, it was found that the left and right side had no significant similarity to one another. Ashraf et al. reported a consistent, slight dissimilarity between the main arteries of the left and right antebrachia overall, with the right side consistently possessing arteries with a slightly larger bore [11]. We found a similar result; however, the larger laterality was not consistently the right (S1 Table). As a consequence, we ran simple linear regression to study the relationship between a single laterality's arterial diameter on the contralateral side. None of the regressions proved predictive ($r^2$ in order: RA 0.098, UA 0.005, SPA 0.062, DPA 0.006). This may demonstrate a complete individuality of the arterial diameters relative to the contralateral side. Several reasons may explain, either in isolation or acting together, the complete dissimilarity between the arterial diameter of each laterality. First, vascular development is known to occur in each extremity independent of the other [23]. Further, the present study examined only external arterial diameter, and did not take any clinical findings into account. Findings that may have caused regular alterations in arterial diameter would include asymmetric limb hypertension, asymmetric vessel atherosclerosis, proximal arterial differences (i.e., subclavian or brachial bore size), and the favored limb of the individual in question. These authors opted not to study the left PMA's effect on the contralateral side, or vice versa, due to an extremely small sample size; these samples would have to include limbs possessing bilateral PMA, which happened in only 4 cadavers with aPMA and 2 cadavers with pPMA.

RA and UA diameter were found to be statistically unchanged with and without the presence of a PMA of any type, however, consistent trends were noted. A slight increase in diameter of the SPA with the presence of a pPMA was noted when examined bilaterally (2.02 mm without, 2.47 mm with pPMA, $p = 0.03$), but neither left nor right extremities alone demonstrated a similar change. As a pPMA often anastomoses with the SPA (5% of cadavers), the increase in size of the SPA does make logical sense due to an additional vascular source [5]. To these authors, it would also make logical sense for the presence of an additional artery in the antebrachium to decrease the sizes of the remaining RA and UA due to an additional downstream pathway for blood flow to the palm. However, this was not found to be the case. In the presence of a pPMA, the radial artery was slightly smaller in diameter than without a pPMA (3.12 mm vs. 3.07 mm, respectively); however, this was not found to be significant ($p = 0.82$). Further breakdown into left and right laterality also demonstrated no significant effect of pPMA on radial artery diameter, though a similar, non-significant downward trend was noted (S2 Table). Interestingly, similar analysis of the UA demonstrated the opposite effect. In the presence of a pPMA, the UA trended slightly larger (3.02 mm with pPMA, 2.75 mm without pPMA; $p = 0.08$). This result was much closer to achieving statistical significance and was exactly the opposite of what was expected. The reason for this enlarging ulnar artery is

not known; these authors invite the reader to ponder the reason this artery may enlarge with PMA presence. One explanation may be an effect of arterial steal, where decreased resistance to flow through a larger vessel increases said flow. The PMA most commonly branches from the ulnar artery itself or from a proximal branch, the common interosseous artery [24,25]. It is conceivable that an additional pathway branching from this artery increases the ability of blood to flow through it rather than through the radial artery, resulting in a larger ulnar artery and a smaller radial artery.

## Conclusion

This and other studies have demonstrated the prevalence of this persisting embryologic artery to be quite high. The present study is the first, to the knowledge of the authors, to examine the external diameter of the persistent median artery and its relationship to the remaining antebrachial and palmar vasculature. Further, there may be clinical implications of altered vascular patterns, potentially impacting orthopedic, plastic, and cardiothoracic surgery. A deep understanding of the variations in vascular anatomy is crucial when surgical procedures are being planned or conducted. For further reading on palmar variants, the interested reader is pointed towards a recent study of the vascular patterns of the SPA done in beautifully cast dissected palms [18]. An understanding of the characteristics and significance of this artery in the adult population is necessary for surgeons operating on the carpal tunnel or distal antebrachium. Further, an understanding of persistent embryologic patterns may point to areas of morphological change over time [4]. Further study of this important artery may impact patient outcomes both operatively and clinically.

## Supporting information

**S1 Table. Arterial diameter measurements in the presence or absence of a persistent median artery (PMA) and its subtypes.**
(DOCX)

**S2 Table. Linear regression analysis demonstrates no correlation between persistent median artery (PMA) diameter and antebrachial artery diameter.**
(DOCX)

**S3 Table. Raw data.**
(DOCX)

## Acknowledgments

The authors sincerely thank those who donated their bodies to science so that anatomical research could be performed. Results from such research can potentially increase mankind's overall knowledge that can then improve patient care. Therefore, these donors and their families deserve our highest gratitude.

## Author contributions

**Conceptualization:** Connor Ellis, Drew Thibault.

**Data curation:** Connor Ellis, Drew Thibault, Josh Lencke.

**Formal analysis:** Connor Ellis.

**Investigation:** Connor Ellis, Drew Thibault, Josh Lencke, Laurieanne D. Hemric.

**Methodology:** Connor Ellis, Laurieanne D. Hemric.

Project administration: Laurieanne D. Hemric.

Supervision: Laurieanne D. Hemric.

Visualization: Connor Ellis.

Writing – original draft: Connor Ellis.

Writing – review & editing: Connor Ellis, Drew Thibault, Josh Lencke, Laurieanne D. Hemric.

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
