## [Editor Report · Decision Letter 0]

24 Jan 2025

PONE-D-24-39864Prevalence and Anatomical Significance of the Persistent Median Artery: A Cadaveric StudyPLOS ONE

Dear Dr. Ellis,

Thank you for submitting your manuscript to PLOS ONE. After careful consideration, we feel that it has merit but does not fully meet PLOS ONE’s publication criteria as it currently stands. Therefore, we invite you to submit a revised version of the manuscript that addresses the points raised during the review process.

We look forward to receiving your revised manuscript.

Kind regards,

Bryan Hozack

Academic Editor

PLOS ONE

Additional Editor Comments:

This study involved extensive cadaveric work and provides interesting information for surgeons and specialists who work in the upper extremity. I think it is important to emphasize in the discussion the fact that there is no clinical information, medical history, or knowledge of past trauma for these patients being studied. There are a wide range of conditions that can affect the vascular anatomy, especially artery thickness, in older adults such as these specimens.

It would be helpful to have a more clearly labeled anatomic image to identify a good example of a persistent median nerve and also the location and manner of measurement of the vessel diameters.

Overall, this could potentially add to our knowledge of anatomic variation in the upper extremity.
---

## [Author Response · Author response to Decision Letter 1]

12 Feb 2025

1. I went through and revised the style of the paper to match the provided templates.

2. I added all the raw data used in this study as “Supplemental Table 3”, keeping all data anonymous/de-identified. Specimens 52-56 were immeasurable d/t preservation, AV fistulae, or other reasons; all data from these specimens is in the table as “n/a” and was not included in the calculations. Data used in figure 2 is present in the figure legend, and therefore a second table including this information was not provided.

3. Citations in the reference list were updated and revised. An issue with the citation manager was fixed.

Add’l editor comments

1. The absence of clinical information, medical history, and knowledge of past trauma was discussed in the “Materials and Methods” section.

2. An anatomic image with labels demonstrating the median nerve, persistent median artery, and other relevant structures was included. The figure legend discusses the labels for each structure; the most important being the palmar-type persistent median artery. This image was updated with increased structure markers and better labeling.

Thank you for your time reviewing this manuscript.

---

## [Editor Report · Decision Letter 1]

18 Feb 2025

Prevalence and Anatomical Significance of the Persistent Median Artery: A Cadaveric Study

PONE-D-24-39864R1

Dear Dr. Connor Ellis,

We’re pleased to inform you that your manuscript has been judged scientifically suitable for publication and will be formally accepted for publication once it meets all outstanding technical requirements.

Kind regards,

Priti Chaudhary, M.S.

Academic Editor

PLOS ONE
---

## [Editor Report · Acceptance letter]

PONE-D-24-39864R1

PLOS ONE

Dear Dr. Ellis,

I'm pleased to inform you that your manuscript has been deemed suitable for publication in PLOS ONE. Congratulations! Your manuscript is now being handed over to our production team.

Kind regards,

on behalf of

Dr. Priti Chaudhary

Academic Editor

PLOS ONE